# Exploring Novel Biomarkers for an Acute Coronary Syndrome Diagnosis Utilizing Plasma Metabolomics

**DOI:** 10.3390/ijms25126674

**Published:** 2024-06-18

**Authors:** Masayuki Shibata, Masahiro Sugimoto, Norikazu Watanabe, Atsuo Namiki

**Affiliations:** 1Division of Cardiology, Kanto Rosai Hospital, Kawasaki 211-8510, Japan; 2Institute for Advanced Biosciences, Keio University, Tsuruoka 997-0052, Japan

**Keywords:** acute coronary syndrome, biomarker, metabolomics, multiple logistic regression

## Abstract

Acute coronary syndrome (ACS) is a life-threatening condition that requires a prompt diagnosis and therapeutic intervention. Although serum troponin I and creatinine kinase-MB (CK-MB) are established biomarkers for ACS, reaching diagnostic values for ACS may take several hours. In this study, we attempted to explore novel biomarkers for ACS with higher sensitivity than that of troponin I and CK-MB. The metabolomic profiles of 18 patients with ACS upon hospital arrival and those of the age-matched control (HC) group of 24 healthy volunteers were analyzed using liquid chromatography time-of-flight mass spectrometry. Volcano plots showed 24 metabolites whose concentrations differed significantly between the ACS and HC groups. Using these data, we developed a multiple logistic regression model for the ACS diagnosis, in which lysine, isocitrate, and tryptophan were selected as minimum-independent metabolites. The area under the receiver operating characteristic curve value for discriminating ACS from HC was 1.00 (95% confidence interval [CI]: 1.00–1.00). In contrast, those for troponin I and CK-MB were 0.917 (95% confidence interval [CI]: 0.812–1.00) and 0.988 (95% CI: 0.966–1.00), respectively. This study showed the potential for combining three plasma metabolites to discriminate ACS from HC with a higher sensitivity than troponin I and CK-MB.

## 1. Introduction

Acute coronary syndrome (ACS) is defined as a condition in which myocardial ischemia or infarction develops due to the acute occlusion of coronary blood flow [1] induced by the rupture or erosion of atherosclerotic lesion plaques or coronary artery spasm (CAS) [2]. Although marked progress has been made in the therapeutic procedures for ACS, especially percutaneous coronary intervention (PCI), ACS remains one of the leading causes of mortality worldwide [3]. However, prompt treatment of ACS, including PCI or coronary artery bypass grafting, can reduce mortality [4]. Therefore, saving the lives of patients with ACS is highly dependent on a rapid diagnosis.

Generally, ACS is diagnosed based on symptoms such as prolonged chest oppression and pain, abnormal electrocardiogram (ECG) findings, and blood biochemical tests, including troponin T/I and CK-MB. Of these, troponin T/I reflects ischemia-induced myocardial injury with a high specificity, and its concentration in the blood promptly increases (usually within two to four hours) after its onset [5]. Therefore, the decision to perform coronary intervention for ACS-suspected patients is often based on the troponin T/I testing results [5,6]. Indeed, the diagnostic ability is well established and highly recommended for diagnosing patients suspected of having ACS [6]. However, a positive result on a troponin test, even a highly sensitive one, is frequently indefinite or absent in ACS cases [7] and requires serial measurements to diagnose or rule out ACS in some populations [6]. Therefore, developing novel biomarkers more sensitive to ACS is necessary for a more prompt and precise diagnosis of ACS.

Metabolomics is an omics approach that enables the simultaneous identification and quantification of hundreds of metabolites in biological samples. This technology has recently been applied to discover biomarkers for cardiovascular disease [8,9]. It involves systematically analyzing the comprehensive metabolites of biological samples, such as biofluids, cells, tissues, or organs, to elucidate their profile features. The chemo-material and quantitative determination of many metabolites in various biological fluid specimens can provide valuable biochemical information on the state of organisms and the interrelations between the different metabolic processes that define the state [10]. Recent technological advances have enabled the high-throughput profiling of many metabolites in biological samples, increasing their application to disease-specific biomarker discovery, including cardiovascular diseases [11]. The biomarker discovery of cardiac disease has undergone a paradigm shift with recent NMR and MS technology innovations. These new methods enable us to understand that the complexity of metabolic networks can be comprehensively captured rather than the monitoring of an individual metabolic enzyme reaction. These tools can perform the high-throughput screening of various metabolites with a high sensitivity. The acquired metabolic profiles can fill in the missing information in other omics, such as proteomics, and metabolic flux can be monitored using stable isotope methods [11]. Laborde et al. investigated 5-OH-tryptophan, 2-OH-butyric acid, and 3-OH-butyric acid as biomarkers in the plasma samples from ACS patients by using GC-MS. The pattern of these biomarkers may reflect oxidative stress and hypoxia in cardiomyocytes [12]. Würtz et al. conducted a biomarker exploration for cardiovascular disease in a large cohort study of over 7000 subjects and validation in two other cohorts of over 2000 and 3000 subjects. Phenylalanine and monounsaturated fatty acids increased the risk, while omega-6 fatty acids and docosahexaenoic acid decreased the risk of cardiovascular diseases [13]. None of the studies exceeded the accuracy of the existing markers, such as troponin as a single marker, and a panel of markers was used to calculate the disease risk, considering each individual’s diversity.

Therefore, we first attempted to elucidate the metabolomics profile characteristics of ACS patients’ plasma in this study. We then explored novel biomarkers of ACS using plasma metabolomics by applying a multiple logistic regression (MLR) model that is more sensitive than troponin I and CK-MB.

## 2. Results

### 2.1. Clinical Characteristics of the ACS and HC Groups

The ACS group (*n* = 18) consisted of patients with ST-elevated myocardial infarction (STEMI; *n* = 14), non-ST-elevated myocardial infarction (NSTEMI; *n* = 2), and CAS (*n*= 2). Patients with STEMI and NSTEMI underwent PCI using conventional balloon angioplasty in one case and drug-eluting stent implantation in the other. All the patients were categorized into Killip class I. In addition, no major adverse cardiovascular events were observed during the hospital stay in the ACS group. The HC group consisted of 24 individuals age-matched to the ACS group. The clinical and laboratory data for both groups are listed in Table 1. The mean age was not significantly different between the two groups. Blood pressure and heart rate were significantly higher in the ACS group, and the prevalence of current smoking was higher in the ACS group than in the HC group. Blood cell counts and biochemical examinations showed that the white blood cell count, creatine kinase, and blood sugar levels were significantly higher in the ACS group than in the HC group. Conversely, high-density lipoprotein-cholesterol levels were significantly lower in the ACS group than in the HC group. No other history of the disease was observed in either of the groups.

### 2.2. The Comparison of the Metabolomic Profiles among the ACS Patients and HC Controls

The metabolomic analyses analyzed 145 metabolites, and the 75 most frequently detected (>90% of samples) were used for subsequent analyses. Figure 1 shows the overall metabolomic profiles of the plasma samples obtained from the ACS patients soon after their arrival at the hospital (ACS group), those obtained from the ACS patients on the day of discharge (pACS group), and those obtained from healthy controls (HC group). The heatmap shows the uniqueness of the ACS group samples, which differed from those of the other groups (Figure 1A). The metabolites in the upper half of the heatmap showed a decreased concentration in the ACS group compared to the other two groups, whereas the bottom half showed the opposite trend. The PLS-DA results differed among the three groups (Figure 1B). Compared to the HC and pACS groups, the plots in the ACS group were scattered over a larger space, which indicated that the metabolomic profiles of ACS showed large individual variations compared to those in the other groups, suggesting that the subjects in the ACS group suffered from various dysregulations of their metabolic status, such as myocardial injury. The metabolites with a high variable importance in the projection (VIP) score obtained via the PLS-DA contributed to the differences among the three groups (Figure 1C). 2-Hydroxyglutarate, lysine, citrate, tryptophan, lactate, isocitrate, creatinine, cis-aconitate, serine, and carnitine showed relatively high VIP scores (VIP > 1.5).

### 2.3. Differences in Metabolites between the ACS and HC Groups

The metabolites showing differences between the ACS and HC groups were visualized in volcano plots (Figure 2). The 12 red-colored metabolites were significantly higher in the ACS group. The 12 blue-colored metabolites were significantly lower in the ACS group than in the HC group (false discovery rate [FDR]-corrected Mann–Whitney test).

Enrichment (Figure 3A) and pathway analyses (Figure 3B) were conducted to understand the pathway-level differences between the ACS and HC groups. Both analyses ranked biotin metabolism and lysine degradation at the top. However, their pathway impacts were small (Figure 3B). Only one metabolite was mapped to biotin metabolism. Four metabolites were mapped to the lysine degradation pathway, comprising 25 metabolites. d-glutamine and d-glutamate metabolism pathways were ranked third and showed a relatively high pathway impact because three metabolites were mapped to the metabolism, including six metabolites.

### 2.4. An MLR Model as a Novel Biomarker for the ACS Diagnosis 

To evaluate the predictive ability of the combination of multiple metabolites to discriminate ACS from HC, we developed an MLR model. Lysine, isocitrate, and tryptophan were selected as minimum-independent metabolites. The AUC value for discriminating ACS from HC was 1.00 (95% confidence interval [CI]: 1.00–1.00) (Figure 4).

A k-fold CV was conducted to evaluate the generalization ability of the developed MLR model. The mean AUC values of the 200 experiments were 0.991 (k = 10), 0.980 (k = 5), and 0.957 (k = 2), respectively (Appendix A). The coefficients of the MLR model were −10.00 (lysine), 12.00 (isocitrate), and −10.93 (tryptophan), and the intercept was 928.1. The AUC values of CK-MB and troponin I were 0.988 (95% CI: 0.966–1.00) and 0.917 (95% CI: 0.812–1.00) (Figure 4).

The ROC curves of lysine, isocitrate, and tryptophan for discriminating ACS from HC are depicted (Figure 5A). The differences in these metabolite concentrations between the ACS, pACS, and HC groups were visualized (Figure 5B). These three metabolites showed significant differences between the ACS and HC groups (Dunn’s post-test after the Kruskal–Wallis test), whereas there was no significant difference between pACS and HC.

## 3. Discussion

In this study, we first elucidated the dynamics of the plasma metabolomic profiles of patients with ACS compared to HCs using LC-TOFMS. The heatmap shows prominent metabolomic profiles of the ACS group with various metabolite concentration changes. The PLS-DA also revealed a distinctive signature of the ACS group metabolomics profile, showing that the plots of metabolomics were markedly dispersed compared to those of the pACS and HC groups. However, it is noteworthy that the pACS metabolomic plots in the PLS-DA were closer to those of the HC. These data might imply that the treatments performed for ACS subjects, including PCI and anticoagulant administration, played preventive roles in extending myocardial injury and the therapeutic effect on myocardial ischemia. We then explored novel ACS biomarkers that can diagnose ACS with higher sensitivity than those of high-sensitivity troponin I and CK-MB using an MLR model with three metabolites: lysine, isocitrate, and tryptophan. This is the first report to propose a combination of novel ACS biomarkers with higher sensitivity than troponin I and CK-MB.

To date, alterations in the metabolomic profiles due to cardiac ischemia, including ACS, have been reported using various tools for metabolomics analyses. Sabatine et al. first suggested that, by comparing human cardiac ischemia to control subjects before and after exercise stress, plasma metabolomics could be applied to determine the changes in groups of functionally related metabolites, leading to novel insights into the pathophysiological metabolomic features of human myocardial ischemic injury [14]. Using LC-MS, they found that 23 metabolites changed due to cardiac ischemia, and 6 were involved in the citric acid pathway (tricarboxylic acid [TCA] cycle). Following this report, TCA disturbances have been observed in patients with myocardial injury and ACS [12,15,16]. The TCA cycle in mitochondria releases energy as ATP by oxidizing acetyl-CoA, produced from various nutrients, such as carbohydrates, proteins, and lipids. The perturbation of the TCA cycle via a decrease in or cessation of oxygen supply due to severe coronary artery stenosis or occlusion induces myocardial dysfunction and cardiomyocyte death. Therefore, the alteration of the TCA cycle metabolomic profiles is a critical sign of cardiac ischemia and ACS. In addition to TCA cycle perturbation, a reduction in the oxygen supply to the myocardium induces glycogen breakdown and an increase in the glycolysis rate, an ATP anaerobically productive pathway, thereby increasing the lactate production [17,18]. Thus, the alterations in metabolites related to the TCA cycle, such as citrate and isocitrate, and glycolysis, such as lactate, are critical markers of ischemic cardiac injury [17]. In this respect, our current study observed the changes in these key pathways of glycolysis and the TCA cycle (Figure 3A) as well as the changes in the metabolites involved in these pathways, including citrate, isocitrate, and lactate, in the AGS group (Figure 2). These data indicate that the current study used adequate and accurate methods and techniques, as in previous reports.

Besides the perturbation of the myocardial energy supply pathway, the increase in oxidative stress is also a major event in cardiomyocytes during ACS. During myocardial ischemia, increased amounts of reactive oxygen species are produced in cardiomyocytes, leading to cell death [19]. Considering the enrichment (Figure 3A) and pathway analyses (Figure 3B), the relatively high impact of the d-glutamine and d-glutamate metabolism pathways on ACS is essential, according to the present data. Since the d-glutamine and d-glutamate metabolism pathways play crucial roles in producing antioxidants that exert cardioprotective effects during myocardial ischemia [20], the high pathway impact of the d-glutamine and d-glutamate metabolism pathways might imply protective feedback responses in cardiomyocytes against ischemia-induced oxidative stress.

In the present study, we analyzed and compared the metabolomic profiles on the day of discharge from the hospital (pACS group) and at hospital arrival (ACS group). As shown in Figure 1, the pACS metabolomic profiles differed from those of ACS (Figure 1A), closer to those of HC (Figure 1B). In particular, lactate, citrate, and isocitrate, the metabolites of the critical pathways of ischemic cardiac injury, TCA cycle, and glycolysis, were markedly decreased in the pACS group compared to the ACS group. Thus, it is reasonable to speculate that the reduction in the critical metabolites associated with ischemic myocardial injury implies that the treatment of coronary artery stenosis and/or occlusion with PCI and anticoagulants was successful, resulting in all ACS patients enrolled in this study being discharged from the hospital alive.

ACS is generally diagnosed based on the symptoms of chest pain and/or oppression, abnormal ECG findings, and blood biochemical tests, such as troponin T/I and CK-MB. Serum troponin T/I and CK-MB are biomarkers specific to myocardial injury, and the elevation of their serum concentration is the major indicator for deciding to perform emergent catheter coronary angiography to check the indication for PCI. Although troponin T and troponin I have the very similar ability and characteristics in the ACS diagnosis [5], it has been recently reported that troponin I tends to be more sensitive to coronary artery disease and ischemic outcomes than troponin T, and troponin T is more strongly associated with renal dyscunction than Troponin I [21]. We thus applied troponin I in the present study. However, the sensitivities of troponin I and CK-MB were not 100%. For instance, it takes two to four hours after the onset of ACS until the troponin T/I serum concentration reaches a positive diagnostic value [5]. Recently, Ali et al. reported that, in 30 patients with acute myocardial infarction, no elevation in serum troponin concentration was observed within 2 h after the onset of chest pain. In contrast, the serum metabolomic profiles of the patients at the same time point were quite different from those of HCs, suggesting that some blood metabolites can be novel ACS biomarkers able to diagnose ACS earlier than the established biomarkers, including troponin I and CK-MB [22]. We developed an MLR model to combine multiple metabolites as novel ACS biomarkers. Consistent with previous reports [5,22], both serum troponin I and CK-MB levels in ACS patients at hospital arrival showed high sensitivities, with AUCs of 0.917 and 0.988, respectively (Figure 4). However, surprisingly, an MLR model composed of the three plasma metabolites, lysin, isocitrate, and tryptophan, showed 100% sensitivity with an AUC of 1.0 (Figure 4). These data indicate that this MLR model with three plasma metabolites can be an ACS biomarker that can diagnose ACS earlier than troponin I and CK-MB with extremely high sensitivity. The three metabolites were thus selected as MLR model features, constituting an independent and minimum set. These metabolites also showed a high VIP score (VIP > 1.7) (Figure 1C), suggesting the excellent reliability of this model.

Regarding the biochemical characteristics of the three metabolites, isocitrate is involved in the TCA cycle described above. Blood tryptophan levels are decreased in patients with ACS [23]. During ischemic myocardial injury, tryptophan is metabolized in the endothelial cells, and its metabolites affect several cardiac functions, including contractility [24]. Therefore, the decrease in tryptophan levels in the plasma of patients with ACS might reflect the activated tryptophan degradation in the endothelial cells due to ischemic myocardial injuries.

Although the precise role of lysine in myocardial ischemic injury is still unclear, lysine is well known to be decreased in the blood of ACS patients [23,25]. Lysine inhibition modulates nitric oxide synthesis [25,26], and nitric oxide controls the vascular tone as a vasodilator [27]. Therefore, the decreased lysine levels in ACS patients’ plasma might be a phenomenon that protects against ischemic myocardial injury by dilating the arteries to increase the oxygen supply to the injured myocardium via coronary arterial stenosis and occlusion.

This study has several limitations. Although our study was prospective, the number of patients in each group was relatively small, and the study was performed at one institution. Metabolites may be affected by various factors such as diet, tobacco, and administrated drugs, and the relationship with confounding factors should be investigated in the future. Further research is warranted to validate the MLR model by enrolling more patients at multiple institutions. Although the MLR model was more accurate than troponin I, we have not been able to develop a simple measurement system for metabolites in the MLR. These are low-molecular-weight molecules, which makes antibody development difficult. On the other hand, using MS requires about a day for sample preparation, measurement, and data analysis. Developing a method that allows measurement at a higher speed and lower cost is necessary.

This study explored novel biomarkers for ACS with higher sensitivity than the established biomarkers, troponin I and CK-MB. LC-MS was used for the metabolomic profiles of ACS patients, post-ACS, and healthy controls. The study identified three plasma metabolites—lysine, isocitrate, and tryptophan—that could discriminate ACS from healthy controls more effectively than troponin I and CK-MB. The area under the receiver operating characteristic curve value for discriminating ACS from HC was 1.00, indicating excellent diagnostic accuracy.

## 4. Materials and Methods

### 4.1. Subjects

Since the incidence of ACS has consistently increased in patients under 60 years old in Japan [28,29], we focused on laborers who were productive workers under 60 years old. Furthermore, sex differences are critical in metabolomic analysis, and this study enrolled males under 60 years old because their incidence of ACS is higher. Therefore, patients with ACS under 60 years old who were admitted to our hospital were enrolled in this study (ACS group). All the patients underwent an ECG and blood biochemical tests, including high-sensitivity troponin T and creatinine kinase-MB tests and plasma sampling for metabolome analyses soon after arrival at the hospital. ACS was diagnosed based on the symptoms and ECG findings with or without using a positive serum high-sensitivity troponin T test. Emergent coronary angiography catheters were used for all subjects, and ad hoc PCI was performed for all subjects, except for two CAS subjects. Before emergent coronary angiography, 3000 IU of heparin was intravenously administered, and 7000 IU of heparin was added before PCI. All the patients who underwent PCI were treated with dual antiplatelet therapy and statins for plaque stabilization. Aspirin and calcium channel blockers were started in the cases diagnosed with CAS. Beta-blockers and ACE/ARBs were added for patients with tolerable blood pressure.

For the healthy control (HC) group, age-matched males who visited our Health Administration Center for medical checkups were enrolled. As all the subjects in the ACS group were male, only males were enrolled in the HC group. Plasma samples obtained from 9 ACS and 11 HC subjects were analyzed using a different analytical method in another preliminary study with other research purposes [30], while the remaining samples were used in the present study. However, preprocessing, measurement, and data analyses for metabolomics were conducted simultaneously with other samples to eliminate unexpected bias.

### 4.2. Metabolomic Analyses

Venous samples were collected in blood collection tubes containing EDTA-2Na (Venoject II; Terumo, Tokyo, Japan) through a 22-gauge needle set (Safetouch PSV set; Nipro, Osaka, Japan). After centrifugation for 15 min at 1500× *g*, the plasma was stored at −80 °C until metabolomic analyses. The venous blood samples of the ACS group were collected soon after arrival at the hospital and on the morning of discharge after overnight fasting (pACS group samples). The venous blood samples of the HC group were collected in the morning after overnight fasting, and plasma was stored as described above.

The metabolomic analysis was performed as previously described [31], with modifications. First, 10 μL of the plasma sample was mixed with 90 μL of methanol containing 1.5 μM of each standard compound (d8-spermine, d8-spermidine, d6-*N*^1^-acetylspermidine, d3-*N*^1^-acetylspermine, d6-*N*^1^,*N*^8^-diacetylspermidine, d6-*N*^1^,*N*^12^-diacetylspermine, hypoxanthine-^13^C_2_,^15^N, and 1,6-diaminohexane) and 1 μM camphor-10-sulfonic acid. After centrifugation at 20,380× *g* for 10 min at 4 °C, 90 μL of the supernatant was transferred to a fresh tube and vacuum-dried. Third, the sample was mixed with 10 μL of 90% methanol and 190 μL of water and then mixed using a vortex mixer and centrifuged at 20,380× *g* for 10 min at 4 °C. Finally, 1 μL of each supernatant was subjected to liquid chromatography time-of-flight mass spectrometry (LC-TOFMS).

The parameters used for the measurement instruments, Agilent Technologies 1290 Infinity LC system and G6230B TOFMS (Agilent Technologies, Santa Clara, CA, USA), and the processing of raw data using Agilent MassHunter Qualitative Analysis software (version B.08.00; Agilent Technologies), have been described elsewhere [32].

The peak areas were integrated and divided by those of the internal standards to eliminate the fluctuations in the sensitivity of MS. All the integrated peaks were visually checked, and reintegration was conducted if necessary. Most of the peak areas in the linear range were confirmed. Peaks smaller than the lower linearity limit were treated as non-detected peaks. All the samples were processed and measured for six consecutive days. The standard mixture was measured each day. The variation in metabolite intensities was confirmed to be less than 5% to control for the quantification quality. The intensities of internal standards injected into all plasma samples were also monitored, and their variations were also confirmed to be less than 5%. The absolute concentration of all metabolites was calculated.

### 4.3. Data Analyses

The differences between the metabolite concentrations of the ACS and HC groups were evaluated using the Mann–Whitney test. The FDR was used to correct *p*-values, considering the multiple independent tests. The discrimination ability of each metabolite was evaluated using the receiver operating characteristic curve (ROC) and the area under the ROC curve (AUC). The Kruskal–Wallis test and Dunn’s post-test were used to compare the ACS, pACS, and HC groups.

The MetaboAnalyst (ver. 5.0) [33] was used for the following analyses. A heatmap with clustering, partial least squares discriminant analysis (PLS-DA), and volcano plots were used to visualize the overall metabolomic profile. The accuracy of the PLS-DA was evaluated via R^2^ value, and its generalization ability was assessed via the Q^2^ value using five-fold cross-validation (CV). For heatmap visualization, the log_10_ of the metabolite concentration was transformed to a Z-score. Euclidean distance and the Ward method were used to cluster metabolomic profiles. Metabolites showing large variations within the top 25 based on a non-parametric analysis of variance were visualized. For the PLS-DA, the metabolite concentration was translated to the Z-score. Volcano plots were generated using the log_10_ metabolite concentration. The X-axis indicates the log_2_ of fold change in the averaged concentration of ACS/HC. The Y-axis indicates −log_10_ (FDR-corrected *p*-value) of the Mann–Whitney test. No threshold was used for fold change.

Enrichment and pathway analyses were conducted to evaluate the pathway-level differences between the ACS and HC groups. The data were mapped into the Kyoto Encyclopedia and Genes and Genomes (KEGG) pathway maps. No data normalization or scaling was performed for this analysis.

An MLR model was developed to evaluate the combination of multiple metabolites to discriminate between ACS and HC. Stepwise feature selection (feedforward and backward with a cutoff of *p* = 0.05) was used to select the independent minimum metabolites. First, this method chooses a metabolite showing the highest discrimination ability and subsequently adds metabolites that increase the discrimination ability and are also independent of the selected metabolites. The generalization ability of the developed MLR model was determined using k-fold CV (k = 10, 5, and 2) with 200 random values (Appendix A).

### 4.4. Statistical Analyses

Statistical analyses were performed using the SPSS Statistics software program, version 23 (IBM Corp., Armonk, NY, USA). Normally distributed continuous variables were expressed as the mean ± standard deviation (SD). Non-normally distributed continuous variables were expressed as medians and ranges. Categorical variables were expressed as numbers and percentages. Fisher’s exact or unpaired *t*-test was used to compare the ACS and HC groups.

These analyses were conducted using the JMP (Pro 16.0.0; SAS Institute Inc., Cary, NC, USA), GraphPad Prism (v.8.4.3; GraphPad Software, San Diego, CA, USA), and Weka (ver. 3.6.1.4, University of Waikato, Hamilton, New Zealand) software programs.

## Figures and Tables

**Figure 1 ijms-25-06674-f001:**
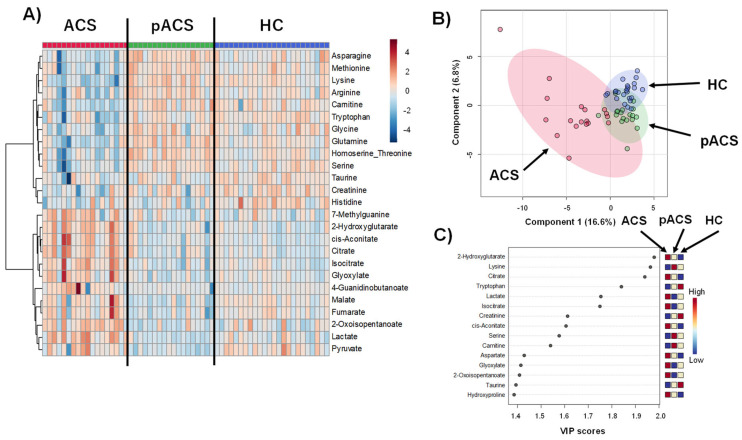
Plasma metabolomics profiles among ACS, pACS, and HC. (**A**) A heatmap of the metabolites showing large variations within the top 25 ranks. Red and blue indicate metabolites with higher and lower concentrations of each metabolite. (**B**) The score plots of the PLS-DA. A plot indicates a sample and 95% confidence intervals are colored. R^2^ = 0.732 and Q^2^ = 0.528 by five-fold CV. (**C**) VIP score of PLS-DA. The metabolites with a higher VIP contribute to the discrimination.

**Figure 2 ijms-25-06674-f002:**
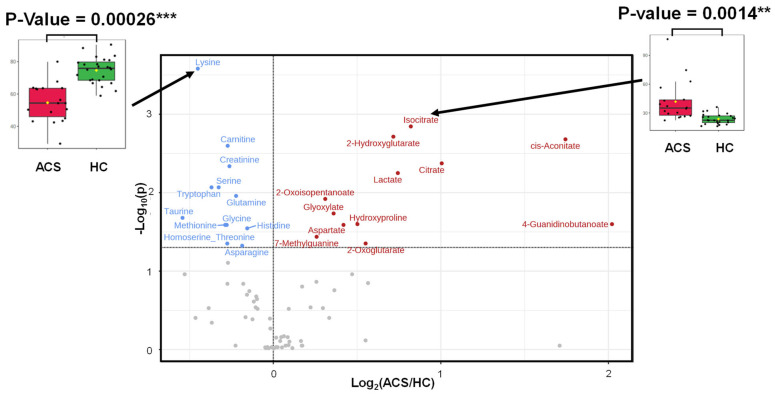
Volcano plots showing the metabolite concentration difference between the ACS and HC groups. The X-axis indicates the log_2_ of ACS/HC (averaged concentration). The Y-axis indicates −log_10_(*p*-value) of the Mann–Whitney test. The *p*-value was corrected by FDR. Y > 1.3 indicates an FDR-corrected *p*-value < 0.05. Two representative metabolites were visualized in the volcano plots. *** *p*-value < 0.0001 and ** *p*-value < 0.01.

**Figure 3 ijms-25-06674-f003:**
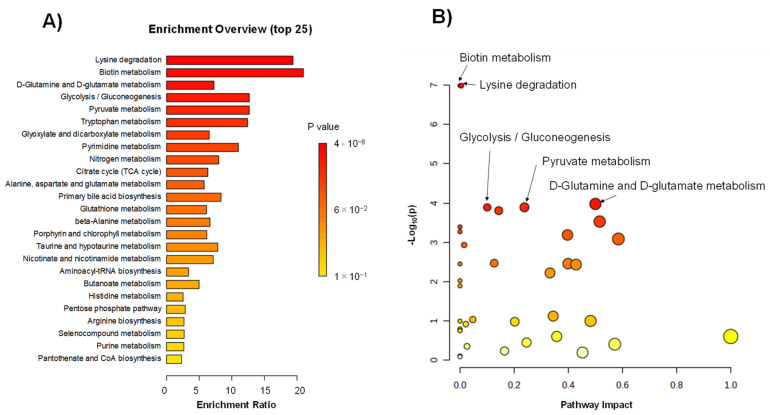
Pathway level difference between the ACS and HC groups. (**A**) The enrichment analysis. The X- and Y-axes indicate the enrichment ratio and *p*-value, respectively. (**B**) The pathway analysis. The X- and Y-axes indicate the pathway impact and −log_10_(*p*-value), respectively. One plot indicates one pathway. Each dot was colored with corresponding *p*-values.

**Figure 4 ijms-25-06674-f004:**
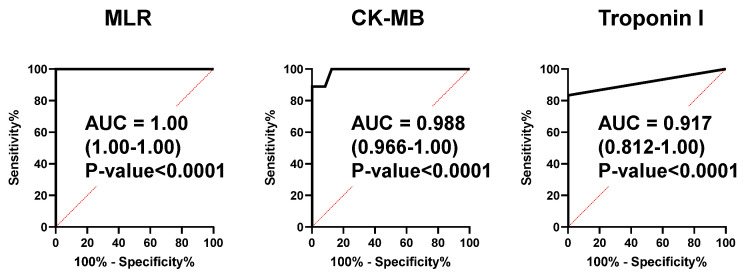
ROC curves to discriminate the ACS and HC groups of the MLR model, CK-MB, and troponin I. The AUC, 95% confidential interval, and *p*-value are presented.

**Figure 5 ijms-25-06674-f005:**
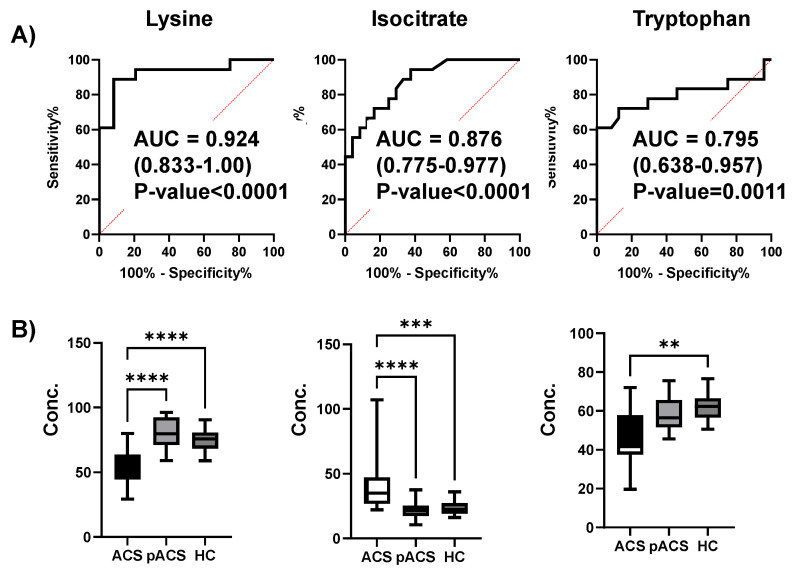
Three metabolites selected by the MLR model. (**A**) ROC curves to discriminate the ACS and HC groups. The AUC, 95% confidential interval, and *p*-value are presented. (**B**) Box plots to show the concentration among the ACS, pACS, and HC groups. Quantitative data were visualized in box plots where horizontal lines indicate 0%, 25%, 50%, 75%, and 100% of the data. Dunn’s post-test after the Kruskal–Wallis test. **** *p*-value < 0.0001, *** *p*-value < 0.001, and ** *p*-value < 0.01.

**Table 1 ijms-25-06674-t001:** Baseline characteristics.

Item	ACS (*n* = 18)	HC (*n* = 24)	*p*-Value
**Age (years)**	49.6 ± 6.6	49.6 ± 7.9	NS
**Body mass index (kg/m^2^)**	23.2 ± 6.3	24.0 ± 2.8	NS
**Blood pressure**			
Systolic(mmHg)	147.4 ± 32.8	117.0 ± 15.1	0.001
Diastolic(mmHg)	102.7 ± 21.3	70 ± 11.5	<0.001
**Heart rate (beats/min)**	70.7 ± 21.9	58 ± 9.3	0.03
**Ejection** **fraction (%)**	53 ± 8.9		
**Comorbidity**			
Hypertension (%)	5.6	4.2	NS
Diabetes mellitus (%)	0	0	NS
Dyslipidemia (%)	5.6	4.2	NS
Hyperuricemia (%)	5.6	4.2	NS
Current smoker (%)	83.3	20.8	<0.001
**Medication**			
ACE/ARB (%)	0	4.2	NS
Calcium-channel blocker (%)	5.6	4.2	NS
Beta-blocker (%)	5.6	0	NS
Statins (%)	5.6	4.2	NS
Xanthine oxidase inhibitor (%)	5.6	4.2	NS
**Laboratory data**			
WBC (cells/µL)	9922.2 ± 3499.0	5525.0 ± 1347.9	<0.001
Total cholesterol (mmol/L)	4.9 ± 0.8	5.5 ± 0.8	0.032
LDL-C (mmol/L)	3.1 ± 0.7	3.5 ± 0.7	NS
HDL-C (mmol/L)	1.2 ± 0.3	1.6 ± 0.3	<0.001
Triglycerides (mmol/L)	5.2 ± 5.3	3.1 ± 1.4	NS
AST (IU/L)	36.3 ± 27.7	24.1 ± 6.7	NS
ALT (IU/L)	33.7 ± 36.3	33.5 ± 20.8	NS
LDH (IU/L)	208.2 ± 91.3	175.0 ± 20.1	NS
γGT (IU/L)	35.9 ± 20.4	44.4 ± 28.2	NS
Sodium (mmol/L)	140.3 ± 1.8	141.6 ± 1.8	0.022
Potassium (mmol/L)	3.8 ± 0.4	4.2 ± 0.3	<0.001
Urea nitrogen (mmol/L)	5.3 ± 1.5	5.1 ± 1.8	NS
Serum creatinine (µmol/L)	79.6 ± 17.7	79.6 ± 8.8	NS
Uric acid (µmol/L)	362.8 ± 113	345 ± 71.4	NS
Creatine kinase (IU/L)	204.3 ± 155.7	114.6 ± 43.3	0.028
Creatine kinase-MB (IU/L)	19.2 ± 18.1		
hs-cTn (pg/mL)	875 ± 1602		
hs-CRP (mg/dL)	0.41 ± 1.21	0.08 ± 0.10	NS
Blood sugar (mmol/L)	153.5 ± 69.1	100.3 ± 7.8	0.006
HbA1c (%)	6.4 ± 2.2	5.5 ± 0.3	NS

Abbreviations: WBC, white blood cell; LDL-C, low-density lipoprotein cholesterol; HDL-C, high-density lipoprotein cholesterol; AST, aspartate aminotransferase; ALT, alanine aminotransferase; LDH, Lactate dehydrogenase; γGT, gamma-glutamyl transpeptidase; hs-cTn, high-sensitivity cardiac troponin I; hs-CRP, high-sensitivity C-reactive protein; NS, Not significant.

## Data Availability

The data presented in this study are available on reasonable request from the corresponding author due to privacy reasons.

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
