# Peer review of "Exploring Novel Biomarkers for an Acute Coronary Syndrome Diagnosis Utilizing Plasma Metabolomics"

_ijms, 2024, doi:10.3390/ijms25126674_

Round 1

Reviewer 1 Report

Comments and Suggestions for Authors

The authors meticulously examined the metabolomic profile of various plasma metabolites, unveiling three novel biomarkers for the diagnosis of acute coronary syndrome (ACS). These proposed biomarkers, Lysine, isocitrate, and tryptophan, exhibit a diagnostic potential that surpasses that of troponin T and CK-MB, offering a heightened sensitivity for the detection of ACS.

The work is innovative and exciting for the clinical diagnosis of ACS.

I'm intrigued by your work and have a few specific questions about it.

1. What other factors can elevate the proposed metabolites which could interfere with your results?

2. The patients studied had some other type of comorbidities that could affect the levels

3. Could you clarify the type of coronary intervention that was performed? Was it consistent across all patients, involving only statins and antiplatelets?

4. The tables need to specify whether the values are percentages (e.g., Hypertension, dyslipidemia, hyperuricemia).

5. Were there inclusion/exclusion criteria in both groups?

6. Why were there only men at the controls? Moreover, in the patients, how many were there?

7. The proposal is exciting and innovative. What advantages could it present over troponin T and CK-MB in terms of ease of use and costs?

Author Response

Reviewer 1
The authors meticulously examined the metabolomic profile of various plasma metabolites, unveiling three novel biomarkers for the diagnosis of acute coronary syndrome (ACS). These proposed biomarkers, Lysine, isocitrate, and tryptophan, exhibit a diagnostic potential that surpasses that of troponin T and CK-MB, offering a heightened sensitivity for the detection of ACS.

The work is innovative and exciting for the clinical diagnosis of ACS.

We thank the reviewer for his encouraging comments.

I'm intrigued by your work and have a few specific questions about it.

  1. What other factors can elevate the proposed metabolites which could interfere you're your results?

We thank the variable comments. The metabolites we studied may be influenced by various environmental factors such as diet and tobacco. Considering the diurnal variation, the time of ACS occurrence and other factors could also be relevant. It is impossible to completely rule out such confounding factors without narrowing the sample of patients at high risk for ACS and rigorous follow-up, and future studies should examine relationships with confounders for which information is available. These issues have been addressed in limitation (L264-265).

  1. The patients studied had some other type of comorbidities that could affect the levels

To clarify, the following sentence was added before Table 1.
"No other history of the disease was observed in either of the groups." (L 88)

  1. Could you clarify the type of coronary intervention that was performed? Was it consistent across all patients, involving only statins and antiplatelets?

Thank you for pointing out. The two coronary spasm cases were continued on aspirin only. The other 16 patients had drug-eluting stents implanted and continued on DAPT. Beta-blockers and ACE/ARBs were added for patients with tolerable blood pressure. We need to consider the impact of these drugs on metabolites in the pACS group. We added these issues to the results and study limitations. (L264-266, L295-297)

  1. The tables need to specify whether the values are percentages (e.g., Hypertension, dyslipidemia, hyperuricemia).

According to the reviewer's comments, we revised Table 1.

  1. Were there inclusion/exclusion criteria in both groups?

Currently, in Japan, myocardial infarction is increasing in the working-age population under the age of 60 (refs. 27, 28). Because sex differences are significant in metabolomic analysis and the incidence is higher in males, this study enrolled males under 60 years of age and age-matched healthy males. We will add this to the materials and methods. (L284-285)

  1. Why were there only men at the controls? Moreover, in the patients, how many were there?

The women were excluded from this study for the reason written in the response to the comments 5.

  1. The proposal is exciting and innovative. What advantages could it present over troponin T and CK-MB in terms of ease of use and costs?

We thank the variable comments. The measurement of the metabolites in the MLR model needs measurement time and cost if we still use mass spectrometry. The development of other assays is necessary for clinical use. To clarify, we described this issue in the limitation. (L276-272)

Reviewer 2 Report

Comments and Suggestions for Authors

The manuscript “Exploring novel biomarkers for an acute coronary syndrome (ACS) diagnosis utilizing plasma metabolomics” by Shibata et. al., is nicely drafted and presented. The manuscript focusses on ACS disease as it is a life-threatening condition that requires immediate medical intervention and early diagnosis of ACS is crucial. ACS usually results from the buildup of fatty deposits on the walls of blood vessels that deliver blood, oxygen and nutrients to heart muscles. Early diagnosis can significantly improve a patient's chances of survival and minimize complications. ACS is diagnosed based on symptoms such as prolonged chest oppression and pain, abnormal electrocardiogram (ECG) findings, and blood biochemical tests, including troponin T and CK-MB. This study aims to explore the clinical metabolomics approach on the identification of novel biomarkers for ACS patients so that to improve early diagnosis and clinical treatment strategies. Based on the findings of LC-MS based metabolomics study, authors claim that this study showed the potential of combining three plasma metabolites to discriminate ACS from HC with a higher sensitivity than that of troponin T and CK-MB.

Comments and suggestions: There are few aspects which need to be addressed to further demonstrate the clinical relevance of this study.

1.      The content of the Introduction is not sufficient, and only briefly introduces ACS and metabolomics. The author should also explain the current research progress of ACS markers, or the application status of metabolomics in ACS, so as to highlight the necessity of this study.

2.      Page 12, line 318: The authors’ stated that normalization and scaling of data was not done. Why is it so? Since normalization is an important part of data processing to reduce the variations from instrument and batch effects.

3.      Figure 1B: What was the criteria followed for the outlier samples. Were they removed?  

4.      Figure 2: The cutoff for Fold change was not marked in the figure. What was the cutoff considered for selecting significant metabolites? Also, the p-value is not shown for the two representative metabolites: lysine and isocitrate. Please take care of the figure labeling.

5.      Page 7, line 122: what was the preliminary criteria for selecting lysine, isocitrate and tryptophan for MLR model?

6.      Page 6, line 104-105: The word ‘p-value’ should be same throughout the manuscript.  

7.      Figure 1: The word ‘Figure 1’ can be removed and the font sizes of title of x and y axes should be increased to become readable.

8.      Conclusion: The conclusion is not standardized, should be the form of full text summary, not just a simple two points of experimental results.

9.      The plagiarism rate is 27%. Please make it below the accepted limit for publication.

10.  Table 1: Please change the fonts and formatting as per the journal’s guidelines.

11.  References:  The references used are very old. Please include the recent references from 2020 to 2024.

The quality of the figures is poor. I recommend this for publication in IJMS after major revisions. 

Comments on the Quality of English Language

English language and Grammar: The writing quality of the paper is poor, and the author needs to revise it greatly.

Author Response

Comments and suggestions: There are few aspects which need to be addressed to further demonstrate the clinical relevance of this study.

  1. The content of the Introduction is not sufficient, and only briefly introduces ACS and metabolomics. The author should also explain the current research progress of ACS markers, or the application status of metabolomics in ACS, so as to highlight the necessity of this study
    According to the reviewer's comments, we revised the Introduction to describe the biomarker discovery studies for ACS. (L51-68)

  2. Page 12, line 318: The authors' stated that normalization and scaling of data was not done. Why is it so? Since normalization is an important part of data processing to reduce the variations from instrument and batch effects.

    We tightly controlled the quantification quality and calculated the absolute concentrations of all metabolites. As pointed out by the reviewer, data normalization is standard without calculating absolute concentration. When we use the MLR model in the clinical setting, we have to measure all metabolites to normalize the quantification levels. The cost and throughput become problematic if we use the current LC-MS system to measure all data. The data analyses using absolute concentration are appropriate to confirm the consistency between the new assay and the result of the current study to develop other assays. We revised the Material, Methods, and Discussion to clarify these issues. (L267-272, L332-336)

  3. Figure 1B: What was the criteria followed for the outlier samples. Were they removed?
    No outlier data were identified, and all samples were used for subsequent analyses.

  4. Figure 2: The cutoff for Fold change was not marked in the figure. What was the cutoff considered for selecting significant metabolites? Also, the p-valueis not shown for the two representative metabolites: lysine and isocitrate. Please take care of the figure labeling.
    We appreciate this comment. These volcano plots were used to visualize all metabolites showing significant differences, and, therefore, the cutoff of fold changes was not visualized. According to the reviewer's comments, p-values were added to two representative metabolites.

  5. Page 7, line 122: what was the preliminary criteria for selecting lysine, isocitrate and tryptophan for MLR model?
    We already wrote the criteria for selecting these metabolites as follows.
    "Stepwise feature selection (feedforward and backward with a cutoff of p=0.05)."
    According to the reviewer's comments, we added the following sentences. "First, this method chooses a metabolite showing the highest discrimination ability and subsequently adds metabolites that increase discrimination ability and are also independent of the selected metabolites." (L361-363)

  6. Page 6, line 104-105: The word 'p-value' should be same throughout the manuscript.

    We thank this observation. We used p-value throughout the manuscript.

  7. Figure 1: The word 'Figure 1' can be removed and the font sizes of title of x and y axes should be increased to become readable.

    The current manuscript deleted these characters in the figures. The font sizes and ases were improved.

  8. Conclusion: The conclusion is not standardized, should be the form of full text summary, not just a simple two points of experimental results.

    According to the reviewer’s comment, we revised the Conclusion. ()L273-279)

  9. The plagiarism rate is 27%. Please make it below the accepted limit for publication.

    We published this paper in Reprint function implemented in this journal. The plagiarism rate of reprints and previous manuscripts might be high. We wrote this manuscript from scratch. However, one possibility of the similarity is the metabolomic protocols written in Material and Methods. We cited the previous report and wrote only the modification to minimize the plagiarism. In addition, the manuscript was revised throughout to improve the English and reduce plagiarism.

  10. Table 1: Please change the fonts and formatting as per the journal's guidelines.

    The table in the current manuscript was formatted according to the journal's guidelines.

  11. References: The references used are very old. Please include the recent references from 2020 to 2024.

According to the reviewer’s comments, we replaced several references with the latest ones. Accordingly, we also revised the Introduction (Ref. 8, 9, and 10, L42-44, L51-68).

  1. The quality of the figures is poor. I recommend this for publication in IJMS after major revisions.

    We improved the resolution of the figures in the current manuscript.

Round 2

Reviewer 2 Report

Comments and Suggestions for Authors

Revision 2

I appreciate the responses given by the authors’, but still some of the points needs to be taken care:

1.      Table 1: The authors’ have modified the table as per journal guidelines. Please state the reason of deleting some characteristics: Ejection fraction (%), hs-CRP (mg/L), hs-cTn (ng/mL). Were they being not important?

2.      Figure 1B: As stated before, one sample in ACS group is seen to be outlier in PLS-DA. If not PLS-DA, what other method used to confirm that no outlier was detected.

3.      Figure 2: The authors’ analyzed the data using software which selects the deregulated metabolites using some cutoff values for fold change and p-value. Please mention the cutoff value of fold change (Log2(ACS/HC)) in the method section.

4.      Figure 1C: The figure needs modification. The word ‘ACS’ is not visible.

I recommend this for publication in IJMS after minor revisions. 

Comments on the Quality of English Language

Grammar and spelling mistakes need to corrected. For e.g. Table1 abbreviations: NS: ‘no sigificicance’ should be changed to ‘not significant’.

Author Response

  1. Table 1: The authors’ have modified the table as per journal guidelines. Please state the reason of deleting some characteristics: Ejection fraction (%), hs-CRP (mg/L), hs-cTn (ng/mL). Were they being not important?

We thank this observation. We revised Table 1 to desrribe the unit.

2. Figure 1B: As stated before, one sample in ACS group is seen to be outlier in PLS-DA. If not PLS-DA, what other method used to confirm that no outlier was detected.

We thank the reviewer’s comment again. The circle indicates 95% confidence intervals and considering the number of ACS, a few plots can be located outside of this circle. Score plots of principal component analyses are sometimes used to detect outlier data. Still, this PLS-DA is a supervised method, and its score plots are usually not used for outlier detection. Some studies use Grubbs' and Dixon's Q tests for outlier detection, but they are parametric tests. Outlier detection is not our purpose, and therefore, the subsequent analyses were conducted with a non-parametric test.

  1. Figure 2: The authors’ analyzed the data using software which selects the deregulated metabolites using some cutoff values for fold change and p-value. Please mention the cutoff value of fold change (Log2(ACS/HC)) in the method section.

    We revised the material and methods to mention that there was no use of cutoff for fold change (L355-L357).

  2. Figure 1C: The figure needs modification. The word ‘ACS’ is not visible.

    We thank again. Figure 1C was edited to show ACS.
  3. Grammar and spelling mistakes need to corrected. For e.g. Table1 abbreviations: NS: ‘no sigificicance’ should be changed to ‘not significant’.

    We revised the manuscript to remove grammatical errors according to the reviewer's comments.